# Sub-Saharan Irregular Migrant Women’s Sexuality: A Qualitative Study in Humanitarian Reception Centers

**DOI:** 10.3390/healthcare12111068

**Published:** 2024-05-24

**Authors:** Alicia García-León, José Granero-Molina, María del Mar Jiménez-Lasserrotte

**Affiliations:** 1Faculty of Health Sciences, University of Almería, 04120 Almería, Spain; aligale97@hotmail.es; 2Department of Nursing, Physiotherapy and Medicine, University of Almería, 04120 Almería, Spain; mjl095@ual.es; 3Facultad de Ciencias de la Salud, Universidad Autónoma de Chile, Santiago 7500000, Chile

**Keywords:** irregular migrant women, qualitative research, reception centers, sexuality, sex education

## Abstract

Irregular female migration to Europe is a growing phenomenon, as more and more women are fleeing their countries of origin due to gender inequality and violence. During the migration process, women experience physical, psychological and social problems that affect their sex lives. The aim of our study is to describe and understand how irregular migrant women living in humanitarian reception centers experience their sexuality at different stages of the migration process. This qualitative phenomenological study collected data through sixteen in-depth interviews with irregular migrant women between January and February 2023. Data analysis was carried out using ATLAS-ti 23.0 software, from which three themes were extracted: (1) The reality of sub-Saharan women’s sexuality, (2) In search of a better life: the choice between taking the risk or surrendering, and (3) The sexual revolution among migrants. Sub-Saharan women’s sexuality is subject to a complex normative order. The migratory process has severe consequences on migrant women’s sex life. The sexual needs of irregular migrant women admitted to humanitarian reception centers undergo a process of change that must be understood by healthcare providers in order to make improvements to care provision.

## 1. Introduction

The migration process is a historical, multidimensional and cross-cultural phenomenon, which profoundly affects women [1]. According to the World Organization for Migration, 281 million migrants moved globally in 2022, while the European Union (EU) received 87 million refugees and migrants [2]. Many of the migrants come from the Middle East, sub-Saharan Africa and Maghreb and have an irregular status when they arrive on the coasts of Spain, Greece and Italy [3]. It is important to differentiate between irregular migrants and refugees as the terms refer to different legal statuses. A refugee has the right to international protection [4] and can be defined as a person who, “owing to well-founded fear of persecution for reasons of race, religion, nationality, membership of a particular social group or political opinions, is outside the country of his nationality and is unable or, owing to such fear, is unwilling to avail himself of the protection of that country” [5]. The term irregular migrant (IM) refers to “a person who, due to unauthorized entry, non-compliance with entry conditions or expiry of their visa, lacks legal status in a transit or host country” [5]. IMs seek protection for various reasons; while some do so on grounds of gender, race, sexuality or religion, others are fleeing war or seeking to improve their living conditions [3]. In recent decades, thousands of IMs have reached Spain by sea in small boats [6]; while they are primarily men, the number of women and children has increased [7]. Of the 40,326 IMs who arrived in Spain by sea in 2020, almost 10% were irregular migrant women (IMW) [8]. It is important to highlight that the socio-cultural context of the IMW’s countries of origin is marked by political oppression, gender inequality, lack of security and a lack of human rights [9]. Besides more general reasons, IMW flee their countries of origin due to violent situations, sexual violence or forced marriages [10]. Migrating from their country of origin via irregular routes is highly dangerous [11] and IMW are a particularly vulnerable group [6]. During the migratory journey, IMW are faced with a precarious situation in terms of food, hygiene and health [3] and are at high risk of violence, sexual exploitation and human trafficking [12,13]. The severity of the journey has a negative impact on their health, affecting their physical, psychological and social well-being [14]. Upon arrival in Spain, IMW receive emergency care. Given their extremely vulnerable conditions, they are admitted to Humanitarian Reception Centers (HRCs) for a maximum of three months, which can be extended [15]. HRCs are part of the Spanish Government’s Humanitarian Care Programme, which attends to the basic needs of vulnerable IMs who reach the Spanish coast. This programme, estimated to hold up to 35,000 migrants in 2023 for an average stay of 64.5 days [16], provides accommodation, food, sanitary facilities, clothing/footwear and basic health care, as well as psychological, social, educational and legal support to IMs [17,18]. When IMW enter HRCs, they undergo a process of adaptation to new social and cultural patterns that affect their physical, psychological, social and sexual health [19,20]. 

The WHO [21] defines sexuality as “a central aspect of human beings, present throughout their lives. It encompasses sex, gender identities and roles, eroticism, pleasure, intimacy, reproduction and sexual orientation”. Understanding social and cultural contexts is fundamental for the care and integration of IMW into host societies [22], as they influence how these women understand their sexuality and exercise their sexual rights [23]. Aboagye et al. [24] note that in most African settings, cultural norms force women to submit to the sexual desires and demands of men, which can lead to situations of violence and pose a health risk for IMW. A lack of social support, fear, low socio-economic status or trauma stemming from the migration journey can influence the sexuality of IMW [25]. Understanding how they experience their sexuality at different stages of the migration process can help improve the sexual and reproductive health of IMW in HRCs. While there are epidemiological [26], clinical [27] and emergency care [10] studies, little is known about how IMW experience their sexuality in the different phases of the migration process. The theoretical framework described by Zimmerman et al. [28] allows us to study the experiences of IMW during the phases of recruitment, transit, exploitation, detention and integration/reintegration, with particular focus on physical, psychological, social and sexual problems or risks. Our data specifically address the beginning of the integration/reintegration phase when the IMW are residing in the HRC; this is a novel approach that adds value to this study. The aim of our study is to describe and understand how IMW living in HRCs experience their sexuality at different stages of the migration process.

## 2. Materials and Methods

This qualitative study used a hermeneutic phenomenological approach guided by the philosophy of Merleau-Ponty [29]. According to this philosophy, we understand the world through our body, focusing on four lifeworld existentials: spatiality, corporeality, temporality and relationality [29,30]. For Merleau-Ponty, the lived body and life experiences must be examined by the participants who describe them [31]. This philosophical perspective was chosen because, while IMW’s sexuality has a social component, they experience violence and violations of their sexual rights, which is more physical in nature. This study followed the COREQ criteria [32].

### 2.1. Participants and Setting

The study took place at HRC facilities for IMW in southern Spain. The participants were selected through purposive sampling. Inclusion criteria included: be an IMW, be ≥18 years old, be of sub-Saharan origin, have arrived in Spain by small boat, have spent at least 2 months in a HRC and have given consent to participate in the study. The exclusion criterion was refusal to participate in the study. For the recruitment of the sample, we were helped by the director of the center and the team of psychologists, who invited the IMW to participate and arranged appointments with them. The collaboration between the researchers and the team of psychologists was crucial in identifying the women who were suited to undergoing the interview, as well as the most appropriate time for data collection. Nineteen participants were contacted, of whom three declined to participate because they did not feel ready to talk about sexuality. The final sample consisted of 16 IMW. The socio-demographic characteristics of the participants can be found in Table 1.

### 2.2. Data Collection

Data collection was conducted through 16 in-depth interviews (IDIs) between January and February 2023. They took place in a classroom at the HRC and had an average duration of 53 min. The IDIs were carried out by two researchers trained in qualitative research. The IDIs used open-ended questions to allow IMW to elaborate on their experiences without the researchers leading the conversation (Table 2). Before starting, the protocol was explained, confidentiality was assured, informed consent was signed, and socio-demographic data were obtained. The IDIs were conducted in English or French, with the help of a cultural mediator and a psychologist from the HRC. The participants’ responses were recorded and transcribed into Spanish. The participants were then given the opportunity to review and approve them. The IMW declined to participate in another interview to clarify and discuss certain aspects of their sexuality in more depth. Data collection was stopped when data saturation was reached.

### 2.3. Data Analysis

The IDIs were analyzed following Colaizzi’s [33] discourse analysis: 1. Familiarization with the data: reading and re-reading of the transcripts to acquire a general understanding of the phenomenon. 2. Identification of statements relevant to the research phenomenon (Quotations). 3. Identification of meanings relevant to the phenomenon that are constructed from the quotations (Coding). An example of the codification process can be seen in Table 3. 4. Search for themes: the identified meanings are grouped into common themes (Categorization). 5. Incorporation of these themes into an exhaustive description of the phenomenon. 6. Creation of statements that capture essential aspects of the phenomenon. 7. Presentation of the statements (step 6) to the participants to ask whether they reflect their experience (validation). Several researchers participated in the review of the transcripts and the categorization. IDIs were recorded, transcribed and analyzed with ATLAS-ti 23.0 software. 

### 2.4. Rigor

Lincoln and Guba’s [34] quality criteria were followed to ensure the rigor of the study. Credibility: triangulation techniques were applied, and the analytical process was independently reviewed by three researchers with experience in qualitative research and immigration. Transferability: a description of the participants, setting, context and method is provided. Reliability and confirmability: a phenomenological approach based on the description of a “lived experience” in a context was used. Verbatim transcripts of the participants’ experiences were reviewed by other researchers, incorporated into the findings through quotations and verified by the participants.

### 2.5. Ethical Considerations 

The study was conducted in accordance with the ethical principles of the Declaration of Helsinki. Approval was obtained from the Almeria Red Cross Committee (protocol code CR_20_01). The participants were asked for written informed consent and permission to record the interviews.

## 3. Results

Three main themes were drawn from the data analysis that describe how IMW living in HRCs experience their sexuality at different stages of the migration process (Table 4).

### 3.1. The Reality of Sub-Saharan Women’s Sexuality

Sub-Saharan women are at a disadvantage due their gender and origin, which can be exacerbated by a lack of social support networks. These factors influence their lives and condition their sexuality. Within this theme, three sub-themes were created.

#### 3.1.1. The Normative Order: Sexual Repression

There are many factors that influence the sexuality of IMW. Africa is characterized by different traditions and cultures and a patriarchal system. Therefore, the sexuality of sub-Saharan women is not a universal and extendable concept, as each woman experiences sexuality in her own unique way. Nonetheless, women’s roles and identities are clearly defined within family, social and work contexts.

“*Over there, women only take care of children, clean and do everything. Men here are equal (to women) but in Nigeria they don’t do anything. In my country it is an abomination for men to cook a meal*”(IDI-1)

The dominant ideology of masculinity, characterized by gender and financial and power differences, has been reflected in the way women experience sexuality and exercise their rights. In a patriarchal society, it is the man who makes decisions about his and his partner’s sex life and reproductive life plans. In addition, the participants revealed that a man’s financial status and privilege even gives him the right to inflict violence on his partner.

“*Yes, it is different here to in my country. For example, there you have rights if you have money. If a man rapes you and he has money, he won’t face repercussions. Over there, if you have money, you have rights. If you don’t have money, you don’t have any thing*”(IDI-2)

“*The man decides whether or not to have more children. Your husband decides whether or not he wants to use contraception… Even when it comes to masturbation, if your partner is next to you, you can*”(IDI-11)

In terms of geographical differences, there are disparities within different communities. Senegalese women stated that they could dress or socialize more freely in the capital than in the villages. Furthermore, sexuality is taboo, and one participant said that women’s parents support these norms, which leads them to have secret relationships.

“*It’s all a bit hidden there. You can’t do anything in front of your parents. In the city, people have more freedom than in the villages*” (IDI-14)

With regard to religion, Christianity and Islam are widely practiced. Although Islam has a positive attitude towards sexuality, society regulates it according to socio-political norms. A patriarchal society structures sexuality around male desire and requires women to be submissive, thus controlling female sexuality. According to the Muslim participants, female virginity is a symbol of honor for the woman and her family, but it evidently implies control over her body. The participants stated that marriage is focused on uniting families, which deprives women from choosing their partner and potentially from finishing their education. 

“*Because of our Muslim religion, we often have to stop studying and get married early, and that also has an impact. We can’t have sex before marriage and in my country, it is not possible to get an abortion as it is strictly forbidden for Muslim women*”(IDI-15)

Tradition and native customs have an enormous influence on the way in which female sexuality is understood. There are taboos and established beliefs that justify many prevalent practices. Women are expected to be decent and reputable; they learn how to behave from a young age, and it is impossible for them to establish equal emotional relationships due to the physical and sexual repression to which they are subjected. 

“*In my country, the tradition is that if you want to get married, your father teaches you, just there in your kitchen, how to talk to your husband. We learn from a young age that we have to take care of the family and it’s something we keep on learning*”(IDI-2)

#### 3.1.2. They Make Decisions about My Body, They Control My Sexuality

Sexual rights include equal legal protection, the right to health care, free reproductive choices, safety and comprehensive sex education. In sub-Saharan Africa, there are barriers to accessing basic sexual health services. Sex education is not a policy strategy in these countries, and gynaecological care, which is linked to family finances, is not accessible to all women. Younger IMW are unaware that these services exist or only associate them with antenatal care and screening for infectious diseases. 

“*Yes, yes, but if you have money, otherwise you can’t go. In Nigeria they will charge you a lot of money. Here (Spain), when we get here they do check-ups and there they don’t, just to see if you have AIDS or not*”(IDI-7)

“*When I was in my country, at school I learned about sexuality a little bit, but not much. When you speak English, yes, you learn that (internet), so you don’t get pregnant*”(IDI-6)

Menstruation is considered a taboo subject shrouded in widespread stigma. The participants did not receive information about menstruation and mentioned a complete lack of social support. The Senegalese women reported that the most commonly used materials were sanitary towels, while in Nigeria, cloths are common in rural areas. The Guinean participants noted that it is very hot in Africa, and they always carry a hygiene product, even if they are not menstruating. In addition, there are still myths related to menstruation that are not based on scientific evidence.

“*You can wash yourself but you can’t go for a swim, because it comes out. You can’t when you’re menstruating. When it’s over, the beach, walking…, but not when you have your period, no. In my house in the bathroom yes, but not with others, very ugly*”(IDI-3)

With regard to their right to protect their sexual and reproductive health, there are differences from one country to another. Most of the IMW are aware of and use contraceptive methods on a regular basis. The most common are the contraceptive pill, contraceptive injection and condom. The Muslim Nigerian participants mentioned other more traditional methods such as herbs, which they only used with their husband’s permission.

“*It’s called a clip in my country, I use that. It’s to put inside the vagina, small, to close it. That’s done in Nigeria*”(IDI-7)

The right to reproductive self-determination implies that women are free to decide when and how many children they wish to conceive. The study’s participants were aware of the benefits of spacing pregnancies two to three years apart. However, social and cultural constraints and gender inequality interfere with IMW’s reproductive rights. One participant stressed the importance of communicating their concerns with their partner, so that they can exercise their sexuality and reproductive rights. However, due to gender discrimination, the man makes the final decision, be it to respect or disregard his partner’s choice. 

“*There has to be communication between husband and wife. For example, when I was in my country, I told my husband that I didn’t want to have another child and my husband let me have only two children. Now I want to take good care of my children, I don’t want more children*”(IDI-2)

“*We don’t decide. My uncle told my aunt that he wanted to have 50 children. If the baby is already nine months old, he says he wants more, he wants more. It’s the man who decides*”(IDI-13)

Regarding abortion, most IMW took an anti-abortion stance and pointed out that this practice is forbidden for Muslim women. However, there is a more neutral stance towards abortion in cases where a woman seeks to terminate a pregnancy following rape. 

“*I don’t think it’s good, it’s better to use methods not to get pregnant. You can also have an accident and I can’t judge that. You have to terminate the child, because it is a bad memory*”(IDI-9)

Female genital mutilation (FGM) is still practiced in many sub-Saharan African countries and more than 50% of the participants had undergone it. It is a practice that is linked to cultural values and beliefs that oppress women. The most frequent motive is to control women’s sexual behavior, rooted in ideals of preserving virginity and reducing promiscuity. In Guinea, religion is not a factor, whereas in Côte d’Ivoire, the IMW highlighted that it was only practiced on Muslim women, especially in the north-east of the country. In Nigeria, however, it depends on one’s educational level. All the participants stated that it is a negative practice that violates their rights as women and affects their sexual pleasure. One IMW in the study expressed the physical and psychological problems she suffers as a result of FGM. 

“*An old woman cut it, I didn’t know her. We went to her house. They hurt me because I tried to defend myself. After they did it to me, I got into a fight with someone and they kicked me down there and that made me bleed even more*”(IDI-13)

#### 3.1.3. Sexual Practices: Caught between Variety and Taboo

Sexual autonomy refers to a person’s right to make decisions in relation to their sexuality. The participants were interested in the topic of homosexuality, which is frowned upon and seen as morally wrong in Africa. They pointed out the differences that exist between Spain and their countries of origin, highlighting the freedom of choice they have now. Although some IMW voiced disapproval, the majority were respectful on the matter:

“*I don’t want that, but I can’t judge. For me it is not wrong if it is your heart that chooses*”(IDI-4)

The IMW stressed that in their countries, men seduce women and not the other way around. They claimed that if it were the other way round, the woman would be labelled as a prostitute. They asserted that men have a dominant role in relationships. 

“*In Africa it is the man who seduces, sees you, asks for your phone number and then invites you to have a meal. It is not good to take the initiative because you can be mistaken for being a certain type of woman*”(IDI-16)

In terms of sexual practices, similar patterns linked to culture and tradition are evident. Social, moral and religious factors condition sexual experiences. Most of the participants were interested in issues related to their sexual activity. They stated that they had never been asked about their likes and preferences, and some were unable to talk about it due to the violence they had suffered since childhood. 

“*Since childhood I have been forced to do all the sexual things they wanted. I don’t really know what I like and what I don’t like. That’s why I have a closed heart, very much so*”(IDI-4)

“*Here it is not necessary to be married to have sex, but in my country I could not*”(IDI-13)

Most of the participants did not express their sexual fantasies due to shyness, modesty or simply not having them. The IMW submit to the desires and demands of their partners, do not have a clear concept of what they want, and reflect a subordinate attitude regarding their relationships with their partners:

“*I used to call my husband at the office. I was a teacher and my husband was an accountant. In the morning, with the phone, like this, different styles, … making love by mobile phone*”(IDI-3)

“*May my future husband be clean*”(IDI-14)

Oral sex is part of sexual activity. However, masturbation is not practiced and only seen in videos as it is deemed shameful and a sin in the eyes of God. Some IMW accept it only in the presence of their husbands, while others reject it outright and see it as incompatible with being a mother:

“*Masturbation, I don’t think it’s good. Why do you do it? If you have a man who does it. It’s perversity. When you have a child you can’t be there touching yourself, you have to present a good image to your child*”(IDI-4)

### 3.2. In Search of a Better Life: The Choice between Taking the Risk or Surrendering

IMW are forced to migrate in search of a better life that guarantees their rights and freedom. Most of the participants arrived on Spanish shores with serious physical and psychological consequences of trauma experienced on the migratory journey and in their countries of origin. Migration journeys are difficult and leaves deep scars that affect their health and sexuality. HRCs are indispensable resources to guarantee IMW shelter and support. 

#### 3.2.1. When a Choice Becomes a Necessity

The IMW referred to different reasons for fleeing their countries of origin, with gender-based violence being a common one. Since childhood they have been vulnerable in a domineering home environment. They live in a patriarchal culture with sexist and male-dominant traditions. They are forced to marry the man the family chooses for them, and if they refuse, they are threatened and may suffer consequences. This leads the IMW to flee their homes in search of refuge. One participant reported that she had been forced into marriage by her father when she was a child: 

“*Why did I come here? Because my father wanted to marry me off and told me to stop going to school… that I had to get married and I said no! And he said that if I don’t get married, he’s going to kill me! That’s why I ran away from home*”(IDI-15)

The IMW are victims of violence that takes many forms. Half of the study’s participants were subjected to female genital mutilation as children, a cultural tradition to control their sexuality. Since then, they have suffered physical and psychological consequences that affect their sexual health. As one of the participants put it: 

“*To this day, when I look at my scar, I feel bad and have bad memories about it, it does affect me*”(IDI-10)

Gender inequality, vulnerability and a lack of safety drove the IMW to flee their countries in search of the rights denied to them in their countries of origin for being women. They began a dangerous migratory journey in search of new opportunities in Europe. 

“*I had no choice but to flee. I didn’t know what awaited me during the journey… but I had no support in my country*”(IDI-7)

#### 3.2.2. Save Our Souls

The migration journey is long, costly and can last between months and years. Migratory routes vary depending on the origin but are often clandestine and pass through areas without police controls. During the migration journey, most of the participants were threatened, robbed, bribed and subjected to strict control enforced through violence. They did not feel prepared to recount the traumatic moments they experienced, let alone those related to sexual experiences. They said that the worst moments were at the Moroccan border. One of the participants reported that she was raped: 

“*And the bad thing is lots of things in Morocco, because the police here steal…lots of things. Like the police raped me, everything, yeah*”(IDI-12)

From Morocco, IMW crossed to the Spanish coast in overcrowded, poor quality small boats. Some of them were pregnant or carrying small children and did not have basic resources or access to safety. They claim that the only thing that mattered to them was to arrive safe and sound; their sexuality was a distant memory. They affirmed that there was nothing positive about the journey and that their only concern was to get to the coast:

“*There are no good things on the trip. When the Guardia Civil took me from the water, it was the only good thing of the whole trip*”(IDI-11)

“*The trip was terrible, the weather was bad, we had no life jackets and the boat was not in a good enough condition to reach land, it was a guaranteed death*”(IDI-9)

The care received by NGOs at the HRC was the most fulfilling experience for the IMW. When they arrived, they were physically and psychologically very damaged and received multidisciplinary care. The psychological care they were given helped them to cope with the adverse events they had experienced and to restructure their new life in a positive way. Within these care programs, sexuality is often neglected.

“*This is all new to me. When we arrived here, we were welcomed as if we were family. I had never experienced that before*”(IDI-12)

“*I am now in counselling to help me overcome what I have experienced… I feel fortunate because for the first time I feel safe*”(IDI-4)

### 3.3. The Sexual Revolution among Migrants

Starting a new life in the host country involves changing customs and lifestyle—a personal transformation that affects all aspects of sexuality. The contrast between cultures leads to personal growth, new expectations and hopes for the IMW; they forget the obstacles they faced and the feelings of sadness and uncertainty when they arrived. The IMW held on to their original beliefs while embracing new ones, which helped them to find the freedom they needed to experience their sexuality.

#### 3.3.1. Barriers to Sexual Satisfaction

The HRC has provided them with a home and the possibility to improve their lives in many ways. The situations of violence they have experienced since childhood have physical and psychological consequences that have an impact on their sexual experiences. Many participants have been genitally mutilated, and therefore have discomfort and difficulties in feeling pleasure. The HRC refers them to the hospital for gynaecological care and to alleviate the problems that arise as a consequence of this practice.

“*I am grateful to the centre for the care I received. Now I know my body better, my sexual parts and I am learning to enjoy myself. They have taught me to love myself as I am*”(IDI-8)

The IMW reported that they are afraid to have new relationships due to their traumatic experiences. Experiences of vulnerability and rape led to feelings of anxiety, fear and difficulty in connecting with men. Some of the participants still do not feel psychologically prepared to deal with new sexual experiences. 

“*I don’t feel like it. In my heart there are no feelings now, it’s cold, I don’t feel like doing that. I don’t want men, just as a friend*”(IDI-11)

“*All rapes leave you disturbed… you stop being a person, you stop being a woman who neither feels nor suffers*”(IDI-6)

The HRC holds sex education workshops to improve women’s awareness. They work on issues related to sexual health promotion, pregnancy and disease prevention. Above all, they teach management and negotiation skills on gender issues in order to empower women and enable them to make decisions concerning their relationships. 

“*Men want to satisfy the more physical part, the pleasure … they don’t care how you feel or what you want. Now I am more at ease because I feel able to decide what I want and say it*”(IDI-9)

“*I’ve come to experience new things. I’ve seen a porn movie and I see different positions… things like that*” (IDI-5)

For other IMW, while they have managed to reignite their sexual desire, they have encountered other challenges. The participants refer to rules and structural barriers that make it difficult to have sexual intercourse in their rooms, including the fact that people from outside the HRC are not allowed to enter. They share rooms with other women or family members, which also makes it difficult to have sexual encounters. 

“*My sex drive has decreased because I don’t have privacy, I won’t make love if the girls are in the room next door*”(IDI-16)

#### 3.3.2. A Complete Personal Transformation

The psychological support received at the HRC helped the women to feel more empowered. The IMW feel freer, more confident, and less constrained. They believe that they have more opportunities to make their own decisions, which has led to changes in their personal and sex lives. Eroticism can be defined as fantasies, erotic dreams, and desire. Some IMW express this desire to feel wanted through focusing on their personal appearance. In their new country they feel free to dress as they wish, without feeling the pressure of imposed norms. The IMW emphasized that they liked to feel desired and wanted to move forward: 

“*I like to make myself pretty, to be wanted. In Côte d’Ivoire you can’t dress the way you want, it depends on your community, your religion. My father wouldn’t let me wear make-up, paint my nails or cut my hair. Here I feel free!*”(IDI-11)

Migration has allowed these women to live out hidden fantasies and feelings, which they could not enjoy in their home countries out of fear of reprisals. Some participants expressed their freedom and desire to be able to have sex before marriage and even to have sex with people of the same sex, which is punishable in their countries of origin. 

“*At times I have even had dreams about other women*”(IDI-11)

“*I like being attractive, I like the physical and sentimental side of a man, but now I want the sexual side. Here you don’t need to be married before having a sexual relationship*”(IDI-5)

## 4. Discussion

The aim of this study was to describe and understand how IMW who arrive by boat and live in HRCs experience their sexuality at different stages of the migration process. Our phenomenological approach has allowed us to achieve a deep understanding of the phenomena from the lived experiences of IMW. Contrary to our findings on sub-Saharan women, we found no studies confirming that the patriarchal system, capitalism and religion influence the sexuality of all women [35]. As our participants explained, sex education affects IMW’s behavior, ability to make decisions, and attitude towards violent or discriminatory behavior [36]. The IMW in our study have very little say on decisions related to their home life, their studies, or their choice of partner. This inequality extends to relationships, sexual practices and family planning, as well as access to continued education and sexual and reproductive care [37]. The IMW in our study had limited access to reproductive health services due to their lack of financial resources. In the same vein, studies highlight that women’s access to health care services in the sub-Saharan region reflected economic and gender inequality; they are dependent on their male partners for financial support [38,39]. Regarding sex education received in school, the participants considered it to be insufficient. The more affluent IMW, as reflected in other studies [36], tried to compensate for this lack of information by searching for it on the internet. According to López-Domene et al. [10], IMW suffer sexual abuse and violence at all stages of the migration process (2011) and are clearly at risk of disease and unwanted pregnancies. Hormonal contraception (DMPA-IM) is the most widely used method in sub-Saharan Africa [40]; its prolonged use is related to the women’s access to sexual and reproductive health information. According to our findings, use of a contraceptive method tends to be short-term; less effective traditional methods, such as herbal contraceptives, are more common [41]. Fertility management and abortion are subject to issues of gender, equity, morality, religion and cultural norms [42]. 

For IMW, abortion is considered illegal and socially unacceptable. We agree with other studies that society exerts strong control over sexuality and reproduction; social prejudice influences personal behaviors and provoke anti-abortion sentiments [42]. Sexual and reproductive health policies are imperative for IMW to have access to sex education about taboos surrounding menstruation or female genital mutilation [43]. Access to good sex education is vital for menstrual literacy [44] and the prevention of FGM, a discriminatory ritual that violates human rights [43]. According to other studies [45], patriarchal societies in many African countries control the norms of courtship, communication and sexual practices. IMW find it difficult to deny sex to their partners or to make any reproductive decisions, including regarding the use of contraceptives [46]. The study participants were unclear about their sexual desires, lacked autonomy to make decisions and based their relationships on their partners’ wishes [47]. Our study concurs with Speed and Cragun [48] that masturbation and self-pleasure are rejected in Muslim and Christian communities, as they are linked to sin, shame and guilt. Our participants fled a culture of exploitation and violence [49,50]; they migrated to Europe because they suffered from gender inequality [51], violence or genital mutilation [52]. Other factors included a lack of safety and family support [53] or the desire to escape from the prospect of a forced marriage [3]. The IMW highlighted their harrowing sexual experiences during the journey, in which they were robbed, raped and threatened [10]. Reaching Europe by crossing the Mediterranean Sea is the IMW’s only escape route, but it puts their lives at risk [50]. Upon arrival in the host country, the participants were attended to by emergency teams and referred to an HRC, where they were cared for by NGOs. The care and respect they received led them to feel grateful and safe [53]. Vulnerability, patriarchal culture and sexual abuse during the migration journey have a strong impact on the physical and psychological health of IMW [54], severely affecting their sexuality. The HRC’s multidisciplinary team works to facilitate the integration of IMW and helps them to overcome their trauma and recover their sex life [55]. HRCs schedule workshops on sex education and gender and empowerment strategies for IMW [13]. In the host country, IMW feel freer, safer and more prepared to take charge of their sex lives. They begin to understand that control over their sexuality is an indicator of gender inequality, which has limited their freedom and autonomy [54]. IMW are expected to control their sexual urges, not take the initiative in relationships and avoid premarital sex [56]. After their arrival at the HRC, IMW undergo a transformation; their self-perception improves on a personal, social and sexual level. They feel free to dress how they want, feel desired, and have new sexual fantasies and are encouraged to form new types of relationships with men and women. Implications for practice could include developing initiatives to promote sexual and reproductive health in migrants’ countries of origin. In this regard, health projects that involve international cooperation have already been proven to be a valid tool for researching and improving problems related to sexuality and sexual health among IMW [57].

### Limitations and Future Lines of Research

It is difficult to compare some of the results of this study due to the scarcity of research in this area. The sample of sub-Saharan IMW is not homogeneous; their economic, political and socio-cultural characteristics vary according to their country of origin. The IMW refused to carry out further interviews that would have allowed us to delve deeper into unclear or less-explored issues due to a lack of time. The participants were aged between 18 and 40 years, and they often started sexual relations at a young age in their home countries. The interview script does not incorporate a specific question on previous experiences, which may be a limitation of the study. In order to mitigate bias, future research could consider purposive sampling based on educational level, socio-economic status, place of residence or family. Data on family support during the migration process, educational level and whether they lived in rural/urban settings could not be retrieved. Most women have left the HRC, and it is not possible to contact them, which may be a limitation of the study.

## 5. Conclusions

IMW’s sexuality is subject to a moral, religious, social and financial normative order determined by their country of origin. While this experience is shared by women globally, the reality for sub-Saharan women is much more complex. IMW live in a patriarchal culture of oppression, violence and discrimination that drives them to migrate to Europe in search of a life that guarantees their sexual and reproductive rights and freedom. During their migration journey, they suffer from assault and sexual abuse that negatively affects their sex lives. Upon arrival in Spain, the IMW are referred to an HRC, where they receive multidisciplinary care. This is the beginning of a complete personal transformation that influences their sexuality. The IMW begin to feel more empowered and fight against the standards imposed on them by the patriarchal system. These results can be used to improve the sex education of IMW who enter HRCs, which would allow them to improve how they manage later phases such as settling in and adaptation.

## Figures and Tables

**Table 1 healthcare-12-01068-t001:** Sociodemographic characteristics of the participants (N = 16).

Participants	Age	Country of Origin	Marital Status	Religion	Children	Time in the HRC (Months)	Migration Journey
IDI-1	19	Nigeria	Single	Christian	No	5	Travelled alone
IDI-2	40	Nigeria	Married	Christian	Yes	4	Travelled alone
IDI-3	30	Guinea	Married	Muslim	Yes	12	Travelled alone
IDI-4	36	Ivory Coast	Single	Christian	Yes	10	Travelled alone
IDI-5	20	Ivory Coast	Single	Christian	No	5	Travelled alone
IDI-6	21	Nigeria	Single	Christian	Yes	6	Travelled pregnant
IDI-7	38	Nigeria	Married	Christian	Yes	7	Travelled alone
IDI-8	34	Guinea	Married	Muslim	Yes	8	Travelled alone
IDI-9	28	Senegal	Single	Muslim	No	5	Travelled alone
IDI-10	30	Ivory Coast	Single	Muslim	Yes	7	Travelled alone
IDI-11	36	Ivory Coast	Single	Christian	Yes	12	Travelled with 4-year-old daughter
IDI-12	36	Ivory Coast	Single	Christian	Yes	17	Travelled alone
IDI-13	18	Ivory Coast	Single	Muslim	No	4	Travelled alone
IDI-14	39	Senegal	Single	Muslim	No	4	Travelled alone
IDI-15	38	Ivory Coast	Married	Muslim	No	4	Travelled alone
IDI-16	24	Guinea	Single	Muslim	No	2	Travelled alone

IDI = In-depth Interview.

**Table 2 healthcare-12-01068-t002:** Interview guide.

Stage	Subject	Content/Possible Questions
Introduction	My intention	To learn about irregular migrant women’s experiences of how they live their sexuality.
	Ethical issues	Inform participants about voluntary participation, registration, consent, confidentiality of data and the possibility of withdrawing from the study at any time.
Beginning	Introductory question	Could you tell me about your experience and reasons for coming to Spain?
Development	Conversation guide	What has changed in your life since you have been here? Do you think being a woman is different here in Spain to in your country? Do you choose the person you like? Have you ever touched your body? When you have sex with someone, what practices do you engage in (intercourse, oral, anal…)?
Closing	Final question	Is there anything else you would like to add?
	Appreciation	Thank them for their participation, remind them that their interview will be of great use, and make ourselves available to them.

**Table 3 healthcare-12-01068-t003:** Example of codification process and analysis of quotations.

Quote	Initial Codes	Unit of Meaning	Subtheme	Theme
“The man decides whether or not to have more children. Your husband decides whether or not he wants to use contraception… Even when it comes to masturbation, if your partner is next to you, you can” (IDI-11)	Male control, male decision-making	Patriarchy	The normative order: sexual repression	The reality of sub-Saharan women’s sexuality
“You can wash yourself but you can’t go for a swim, because it comes out. You can’t when you’re menstruating. When it’s over, the beach, walking…, but not when you have your period, no. In my house in the bathroom, yes, but not with others, very ugly” (IDI-3)	Menstrual hygiene, cloths, menstruation information, norms.	Menstruation	They make decisions about my body, they control my sexuality.	

**Table 4 healthcare-12-01068-t004:** Themes, subthemes and units of meaning.

Theme	Subtheme	Units of Meaning
The reality of sub-Saharan women’s sexuality	The normative order: sexual repression	Geographical and economic differences, religion, tradition and culture, patriarchy
	They make decisions about my body, they control my sexuality	Mutilation, gynaecological care, menstruation, contraception, sex education
	Sexual practices: caught between variety and taboo	Homosexuality, oral sex, masturbation, sexual fantasies
In search of a better life: the choice between taking the risk or surrendering	When a choice becomes a necessity	Safety, forced marriage, violence
	Save our souls	Journey, violence, fear, welcome, safety
The sexual revolution among migrants	Barriers to sexual satisfaction A complete personal transformation	Traumatic experiences, FGM, housing, school rules, poor sex education Change, empowerment, freedom, sexual fantasy, feeling wanted

## Data Availability

For confidentiality purposes, the data are in the possession of the author (M.d.M.J.-L.).

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
