# Peer review of "Sub-Saharan Irregular Migrant Women’s Sexuality: A Qualitative Study in Humanitarian Reception Centers"

_healthcare, 2024, doi:10.3390/healthcare12111068_

Round 1
Reviewer 1 Report
Comments and Suggestions for Authors
This study provides important insight into the sexual perceptions/experiences if immigrant women. This is an important and understudied topic.
There are many oddly hyphenated words throughout. (e.g. Page 1, line 33 - mi-grant; abstract, line 15 – wo-men, and many, many more).
Title – Should “title” be included at the end of the title?
Introduction:
In the first paragraph, authors need to specify that the IM to which they are referring, who arrive on the coasts of Spain, Greece, and Italy are from sub-saharan Africa. This is not clear.
Additional information on the impact of immigration to another country is needed, especially for women.
Additional information on African traditions/culture/the fact that it is mostly a patriarchal system is needed. Also, the fact that many different cultures and traditions are present needs to be addressed.
What is the difference between an IM and a refugee?
Authors need to more fully flesh out the importance of understanding the sexuality of IMW – the connection between their sexual health and overall wellbeing is not made clear. What previous research has been done in this specific area?
Materials and Methods:
Authors mention abuse for the first time in this first paragraph. This should be referenced in the introduction if this is one of the focal areas of the article.
Line 71 – Typo with “the” and “this” at the beginning of the sentence.
Line 81 - What is the justification for the inclusion criteria – specifically having spent at least 2 months in a HRC, and arriving in Spain by boat?
Line 81 - What is “legal age?”
Ethical considerations should be described earlier in the methods section.
Authors should identify the risks to participants, including emotional and mental health-related risks (e.g. related to trauma).
Were participants provided an incentive for participating in the study?
Typo in line 92 – Delete the word “a”
Typo in line 93 – “which” should be “with”
Line 101 – Did all of the IMW’s decline to do another interview, or just one? This isn’t clear.
Data Analysis:
Authors should provide more detail related to the coding and categorization process, including examples of statements and their relevant codes, and examples of categorization.
Results:
Did marital status impact responses?
Did women who were not married have family/other support systems? Might that have impacted responses?
Did you ask about previous sexual experience, or was it assumed these women had experience with sex?
What impact might sexual education and experience have had on responses?
Line 341 – Typo with 3.2.2 subtheme and 2.2 Save our souls
Discussion:
Authors should expand on the limitation to the study, and explain further the trauma experienced by the respondents which precluded their desire for additional interviews.
Authors should expand on the connection between the study results and healthcare provision/sex education.
Comments on the Quality of English LanguageMinor English language editing required.
Author Response
Authors thank reviewer 1 for his work.
Reviewer 1 can see our response in attached file

Reviewer 2 Report
Comments and Suggestions for Authors
I find the results interesting, filling a gap in the literature.
I have some suggestions and comments, as follows.
In the introduction, I suggest to the authors to add a paragraph reffering to the novelty of the results.
When describing the sample, it would be interesting to know if the participants were migrating with their families (children or/ and husband), and in this regard I would suggest to the author to add a column in Table 1. I find this aspect even more important, as further information are found in the paper and it will be easier for the reader to follow.
Another characteristics which I think are of great help for the readers are the education level of the respondents and if they lived in rural or urban areas before migrating.
In my oppinion, as there are only four home countries for IMW, a short description of socio-economic conditions (female participation in labour market, education level, mother’s mean age at first birth, etc.) would add value to this study. Maybe some information about the reasons of migrating if they are available would be relevant.
I doubt that all IMW migrate to Europe in search of a life that guarantees their sexual and reproductive rights and freedoms. It is important to see how many of them migrated with their families...
An important theme/sub-theme should be the impact of the cultural diversity in the host country (there are only some remarks in the conclusion section (lines 503-504).
This results are of great importance and this qualitative study may be followed by a survey.
Considering the above remarks, my suggestion is to reconsider the paper after major revision.
Kind regards,
Comments on the Quality of English Language
Some further editing should be done.
Author Response

(The authors gave the same response as above.)

Round 2
Reviewer 1 Report
Comments and Suggestions for Authors
One minor issue to address:
Line 101 – should be greater than or equal to 18, not just greater than 18.
Author Response
The authors are grateful for the reviewer comments.
We have corrected Line 102, and added ≥ 18 years old.
Changes are highlighted in red in main document.

Reviewer 2 Report
Comments and Suggestions for Authors
Dear Editor, dear Authors.
Thank you for sending me this manuscript to review.
I am ok with the paper now. From my point of view, the paper may be published.
Kind regards,
Author Response
The authors are grateful for the reviewer comments
